# Temperature and Daylength Effects on Growth and Floral Initiation in Biennial-Fruiting Blackberry

**Anita Sønsteby** [1],* and **Ola M. Heide** [2]

1  Department of Horticulture, Norwegian Institute of Bioeconomy Research (NIBIO), Nylinna 226, NO-2849 Kapp, Norway
2  Faculty of Environmental Sciences and Natural Resource Management, Norwegian University of Life Sciences, NO-1432 Ås, Norway; ola.heide@nmbu.no
*  Correspondence: anita.sonsteby@nibio.no

**Abstract:** Little is known about the environmental control of growth and flower bud initiation (FBI) in commercial blackberries. We studied the processes in the cultivars 'Lock Ness', 'Ouachita' and 'Sweet Royalla' at 12, 16 and 20 °C in a daylight phytotron under naturally decreasing autumn daylength at Ås, Norway (59°40′ N). Growth rate increased with increasing temperature but was much lower at all temperatures in the erect 'Ouachita' than in the trailing cultivars 'Lock Ness' and 'Sweet Royalla'. In all cultivars, FBI occurred earliest at 16 °C, whereas little or no FBI took place in 'Ouachita' and 'Lock Ness' at 12 °C. Growth cessation was earliest at 16 °C where it occurred in early September in all cultivars, suggesting a critical daylength of approximately 14 h. At variance from earlier statements, FBI started in lateral buds situated several nodes below the apex and progressed in both acropetal and basipetal directions as previously reported for red raspberry. Winter chill at 0 °C enhanced flowering in spring in marginally induced plants of all cultivars except 'Ouachita' grown at 12 °C, which remained vegetative in spring. The results suggest that temperature is as important as daylength for FBI in biennial-fruiting blackberry, and that winter chilling may enhance flowering and yield potential in partially induced plants.

**Keywords:** controlled environment; differentiation; floral primordia; primocane-fruiting blackberry; *Rubus*; temperature





## 1. Introduction

Blackberries are native to much of the cold and temperate regions of North America and Eurasia. While traditionally utilized only as a wild fruit, blackberry has now become an important domesticated fruit crop that has experienced a remarkable expansion worldwide, being grown on an excess of 25,000 ha in 2014 [1,2]. Important reasons for this have been the fast development of new cultivars with higher yields and improved fruit quality, modified production practices and new production regions, as well as a rapid increase in the year-round market demands for fresh fruits [2]. Like other *Rubus* species, the blackberry consists of a perennial root system with biennial shoots (canes), and like the situation in raspberries [3–6], both biennial-fruiting and annual-fruiting types are recognized and cultivated (also referred to as floricane-fruiting and primocane-fruiting, and summer-fruiting and autumn-fruiting types, respectively). Since the cultivated blackberries have originated from interbreeding of several genetically heterogeneous *Rubus* species, they are morphologically and physiologically highly diverse. Therefore, they are not designated any specific scientific name, but are commonly referred to as *Rubus* subgenus *Rubus* Watson [2].

Despite the great economic importance of blackberry production, surprisingly little is known about the environmental control of flower bud initiation (FBI) in the plants [7,8]. Although organogenesis and the successive stages of flower bud differentiation have been described [9,10], hardly anything is known about which environmental factors are

triggering the process. However, a range of flowering phenology studies have been conducted at different locations with varying environmental conditions with the aim of determining the seasonal timing of FBI in the various geographic regions [5,6,9–12]. From such information, some assumptions have been made about the underlying factors that are triggering FBI (see below).

The results of these investigations have also revealed notable differences in the seasonal timing of FBI in biennial-fruiting red raspberries and blackberries. While early stages of FBI in biennial-fruiting red raspberry usually become microscopically visible in early September under natural daylength conditions [5,6,13], it is generally observed much later in the season in biennial-fruiting blackberry. Thus, Waldo [11] in his pioneering investigation, found that 'Himalayan Giant' blackberry initiated floral primordia in January in Maryland and in February in Oregon, while its offspring 'Brainerd' initiated flowers during November in both regions. Similar results with 'Himalayan Giant' were obtained by Robertson [12] in Scotland, while Takeda et al. [10] found that 'Marion' and 'Boysen' trailing blackberry initiated floral primordia in November and December in Oregon, whereas the semi-erect 'Chester Thornless' remained vegetative until spring, as was also the case with 'Hull Thornless' in West Virginia [9]. In contrast, Warmund et al. [14] found that 'Darrow' blackberry initiated flower buds already in September at Clarksville, Arkansas. These results show that under natural field conditions, the FBI of biennial-fruiting blackberry occurs relatively late, taking place in late autumn or early winter, and sometimes not until spring. However, because of the diverse genetic background of cultivated blackberry, cultivars with earlier FBI may also occur [14].

Based on these observations, FBI in biennial-fruiting blackberry is generally thought to be a short day (SD) response [8,10]. However, nothing is known about the flower-inducing effect of temperature and its interaction with photoperiod in the process, even though Takeda et al. [10] concluded that temperature plays a major role in determining the extent of bud differentiation during winter. Since the annual changes in daylength and temperature vary in parallel in nature, their specific effects cannot be determined in such field experiments. Nevertheless, the observed differences in the timing of FBI in red raspberry and blackberry under field conditions suggest that the latter may have shorter critical daylengths and/or lower temperature limits for FBI than reported for raspberries.

As in biennial-fruiting raspberries [6], FBI in biennial-fruiting blackberries is usually associated with growth cessation and dormancy induction. Thus, they require winter chilling after FBI for dormancy release and facilitation of budbreak and flowering in spring [15]. Warmund et al. [14] also reported that the endodormant phase in blackberries appears to be completed in January when growth again can be induced by high temperature. However, the chilling requirement for blackberry cultivars and its relation to flowering is far from adequately known and understood [8,14].

For comparison, it should be mentioned that FBI in the annual-fruiting types of blackberry is less strictly controlled by environmental factors, and takes place more or less freely after a short period of vegetative growth [16,17]. This is analogous to the situation in annual-fruiting raspberries which is less photoperiodic dependent and which flower freely across the entire 6–30 °C temperature range [18,19]. Lopez-Medina et al. [16] also found that organogenesis and flower development stages were identical in the two blackberry types.

Blackberry plants are bulky and fast growing with complex morphological structures that make them inconvenient for experimentation in controlled environments. This is probably an important reason for the limited information available on their FBI. However, challenged to provide more exact information on this important issue, we have carried out a simple experiment with three commercial biennial-fruiting blackberry cultivars under controlled environment conditions. The main objectives of the experiment were firstly to gain some experience with experimentation of the bulky blackberry plants under controlled environment conditions, and secondly to study the effect of relevant temperatures on growth and FBI in commercially available cultivars with contrasting growth characteristics.

## 2. Materials and Methods

### 2.1. Plant Material and Cultivation

Since only a limited number of blackberry cultivars are available, the commercially interesting cultivars 'Lock Ness', 'Ouachita' and 'Sweet Royalla' were used for the experiment. The plants were propagated from adventitious root buds and raised in 3 L pots as single-stem plants in a greenhouse at a minimum temperature of 21 °C in continuous light as described for red raspberry plants [5]. On 3 August 2022, when the plants had developed approximately 15 leaves and an average height of 82, 67 and 85 cm, respectively, for 'Loch Ness', 'Ouachita' and 'Sweet Royalla', the experiment was started in the daylight phytotron of the Norwegian University of Life Sciences at Ås, Norway (59°40′ N, 10°45′ E), lasting for 11 weeks until 19 October. During this experimental period, the plants were exposed to naturally decreasing daylength at constant temperatures of 12, 16 or 20 °C, the range of temperatures considered most relevant. Then, one plant of each treatment was harvested and all lateral buds along the entire length of the main shoot and all lateral shoots were dissected for determination of flowering stages, whereas the remaining five plants were moved directly into a cold store with a temperature of 5 °C. After 2 weeks, the temperature was lowered to 2 °C, and after another 2 weeks to 0.5 °C, at which temperature the plants were overwintered. In the following spring, the overwintered plants were moved outdoors for observation of budburst and flowering performance under natural long day (LD) and temperature conditions at 60° N. Temperatures in the phytotron were controlled to ±1 °C, and a water vapor pressure deficit of 530 Pa was maintained at all temperatures.

### 2.2. Experimental Design and Data Sampling

The experiment was factorial with a split-plot design with temperatures as main plots and cultivars as sub-plots. Each treatment was started with six plants of each cultivar, and the data for each plant were treated as replications in the statistical analyses. During cultivation in the phytotron, plant growth was monitored by weekly observations of shoot height and number of leaves in each plant. After 11 weeks of cultivation at the various temperatures, all buds of a single plant from each condition and cultivar, were dissected under a stereo microscope and examined for determination of their flower development stage. Fresh buds were always used, and the flower development stages were scored according to the scale used by Takeda and Wisniewski [9], where fully vegetative buds were designated stage 1, and buds with early perianth primordia were designated stage 2, and fully differentiated flowers with both stamen and carpel primordia as stage 7. However, after four weeks of growth, when the trailing and fast-growing cultivars had reached a height of about 350 cm and became difficult to measure and handle, all plants at 16 and 20 °C (only at 20 °C in 'Ouachita') had to be decapitated as shown in Figure 1. The uppermost lateral bud below the cut was then allowed to grow out to form a continuation shoot. Flowering performance of the five remaining plants in spring were recorded by observations of the percentage of flowering plants and the number of flowering nodes, as well as the total number of flowers in each plant. Experimental data were subjected to analysis of variance (ANOVA) by standard procedures using a Minitab® Statistical Software program package (Release 17.2.1 MiniTab, MiniTab Inc., State College, PA, USA). Tukey's test was used for the separation of means. Percentage values were always subjected to an arcsine transformation before the ANOVA.

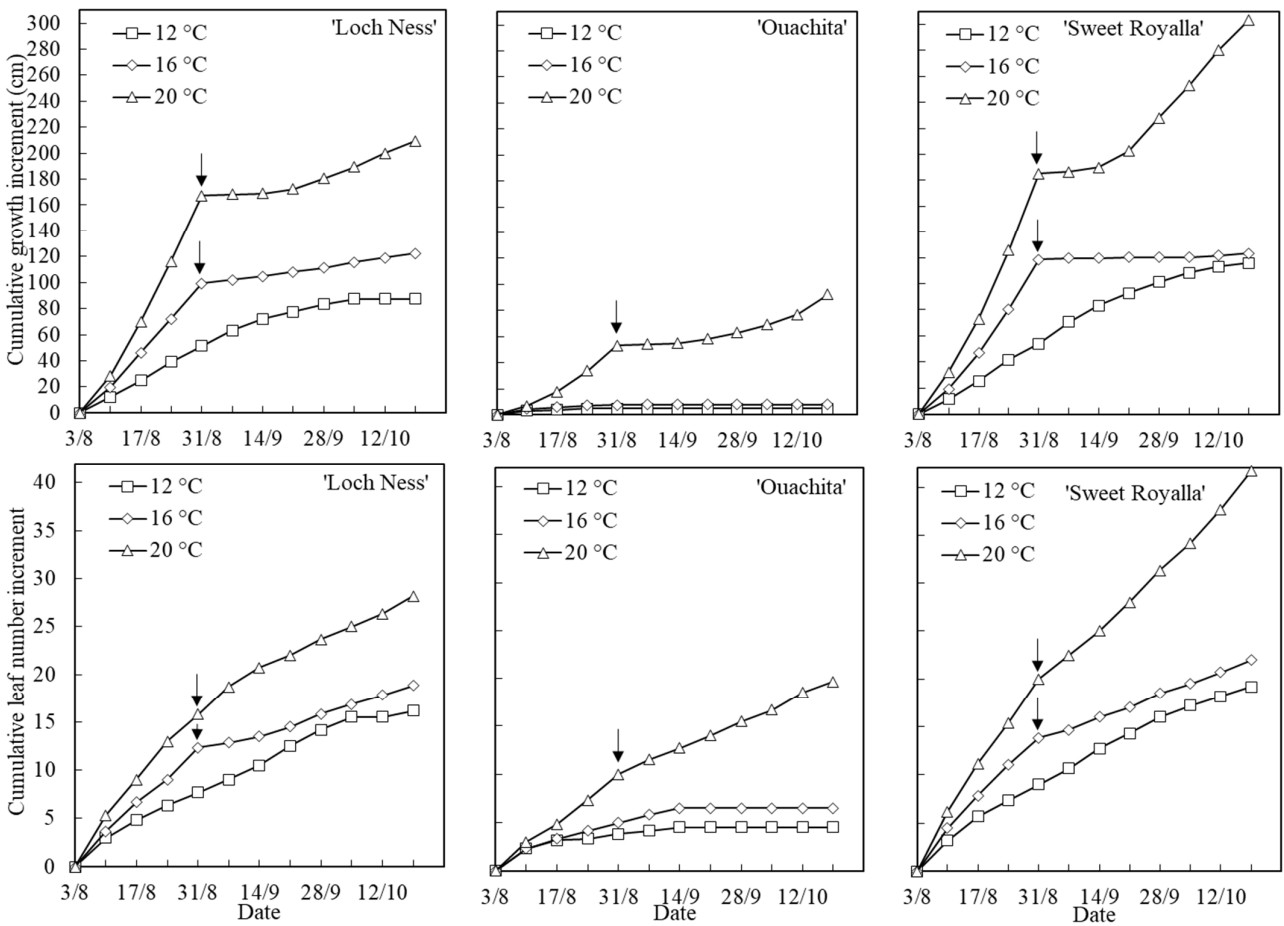

**Figure 1.** Time courses of cumulative shoot growth (above) and leaf number increment (below) in three blackberry cultivars during 11 weeks of cultivation under naturally decreasing daylength at 60° N and temperatures of 12, 16 and 20 °C as indicated. Data are the means of six replicate plants of each cultivar. Arrows indicate time of shoot decapitation.

## 3. Results

### 3.1. Plant Growth

In all cultivars, the trend in both shoot elongation and initiation of new leaves increased significantly with increasing temperature (Figure 1). The rate of growth also varied significantly between the cultivars, being much lower in the erect growing 'Ouachita' than in the trailing 'Lock Ness' and 'Sweet Royalla'. Early cessation of growth took place in 'Ouachita' at both 12 and 16 °C, while only at 12 °C and late in the experimental period in the other cultivars. After decapitation, both shoot elongation and initiation of new leaves were halted for three or more weeks depending on the cultivar's growth vigor. An exception was 'Sweet Royalla' which ceased growing and formed flower buds after decapitation at 16 °C, indicating that the bud subtending the cut had already initiated floral primordia at the time of cutting. However, both leaf production and shoot growth continued at reduced rates after the decapitation in all cultivars. Except for the erect and slow growing 'Ouachita', growth rate of the blackberry cultivars was generally markedly higher than that observed in the genetically related red raspberry under similar conditions [3–5,13].

### 3.2. Flower Bud Formation and Spring Flowering

The flower bud organogenesis and development stages observed in the three blackberry cultivars were identical with those reported by Takeda and Wisniewski [9] and Takeda et al. [10] for a wide range of other biennial-fruiting cultivars. In fully vegetative plants, the apical meristem was small and flat and encircled by leaf primordia only. The first

sign of transition to generative development was a noticeable enlargement and flattening of the meristem followed in fast succession by the formation of three bracts and a whorl of sepal primordia along the apical rim. This was scored as flower development stage 2 and considered the first reliable sign of FBI. Then, the formation of petals, stamens and carpels followed in centripetal order in parallel with enlargement of the central receptacle.

In all cultivars, the most advanced floral primordia were observed at 16 °C (Figure 2). However, while advanced flower buds were observed in 'Sweet Royally' at all temperatures, 'Ouachita' did not initiate floral primordia at 12 °C, despite its early growth cessation at this temperature. This cultivar had advanced floral bud primordia only at 16 °C while at 20 °C, a few marginal floral primordia were observed in a couple of buds on its main shoot only. Likewise, 'Lock Ness' had not initiated floral buds on its main shoot at 12 °C at this time but had a few floral primordia on a basal lateral shoot. However, at 16 °C, and to a lesser extent at 20 °C, this cultivar had advanced lateral floral bud primordia on its main shoot and its basal lateral shoots. On the other hand, the strongly responding 'Sweet Royalla' had initiated abundant and advanced floral buds on both the decapitated main shoots at 16 and 20 °C, and on the non-decapitated main shoot at 12 °C. This was also the case with its continuation shoots at the top of the decapitated plants at 16 and 20 °C, as well as on most of its basal lateral shoots at all temperatures.

It is also interesting to note that the direction of flower development was different in the non-decapitated 'Sweet Royalla' plants at 12 °C and the decapitated plants at 16 and 20 °C. In the non-cut shoots at 12 °C, the most advanced floral primordia were found about 15 to 20 nodes below the apex, with less advanced stages at both higher and lower bud positions. The same was found in non-decapitated plants of 'Ouachita' at 16 °C, except that the most advanced buds there were found at nodes 8–11 below the apex. This was at variance from the decapitated 'Sweet Royalla' plants at 16 and 20 °C, where the shoots were decapitated at the 15–20 node positions, i.e., in the area where the buds were just about to start initiation of floral primordia. This means that the continuation shoots of the decapitated plants at 16 and 20 °C were developmentally comparable to the lower half of the non-decapitated shoot at 12 °C with basipetal direction of flower development.

The green foliage of the plants was retained during cold storage in the dark during winter and during spring growth and flowering. The effects of cultivar and temperature treatment observed by dissection of buds in autumn was generally confirmed by the flowering performance of the plants in spring, although winter chill at 0 °C tended to enhance flowering in spring (Table 1). The advancement of FBI observed at 16 °C in 'Lock Ness' and 'Sweet Royalla' in autumn was reflected in a slight but not significantly earlier anthesis in spring. However, although all plants of these cultivars flowered in spring, the total number of flowers per plant and the percentage of flowering nodes tended to be higher in plants from 12 and 20 °C than in those from 16 °C, although the effect was not always significant. On the other hand, large and significant effects of autumn temperature on spring flowering were observed in 'Ouachita' of which all plants from 12 °C were still completely vegetative, whereas 40% and 100% flowering, respectively, took place in the plants from 16 and 20 °C. However, the number of flowers and the percentage of flowering nodes were always low in this cultivar. Limited shoot growth of this cultivar, and hence the low number of potential flowering sites, was probably an important contributing factor to this poor flowering. Lateral length and the number of flowers per lateral generally increased with increasing autumn temperature with a highly significant cultivar x temperature interaction (Table 1).

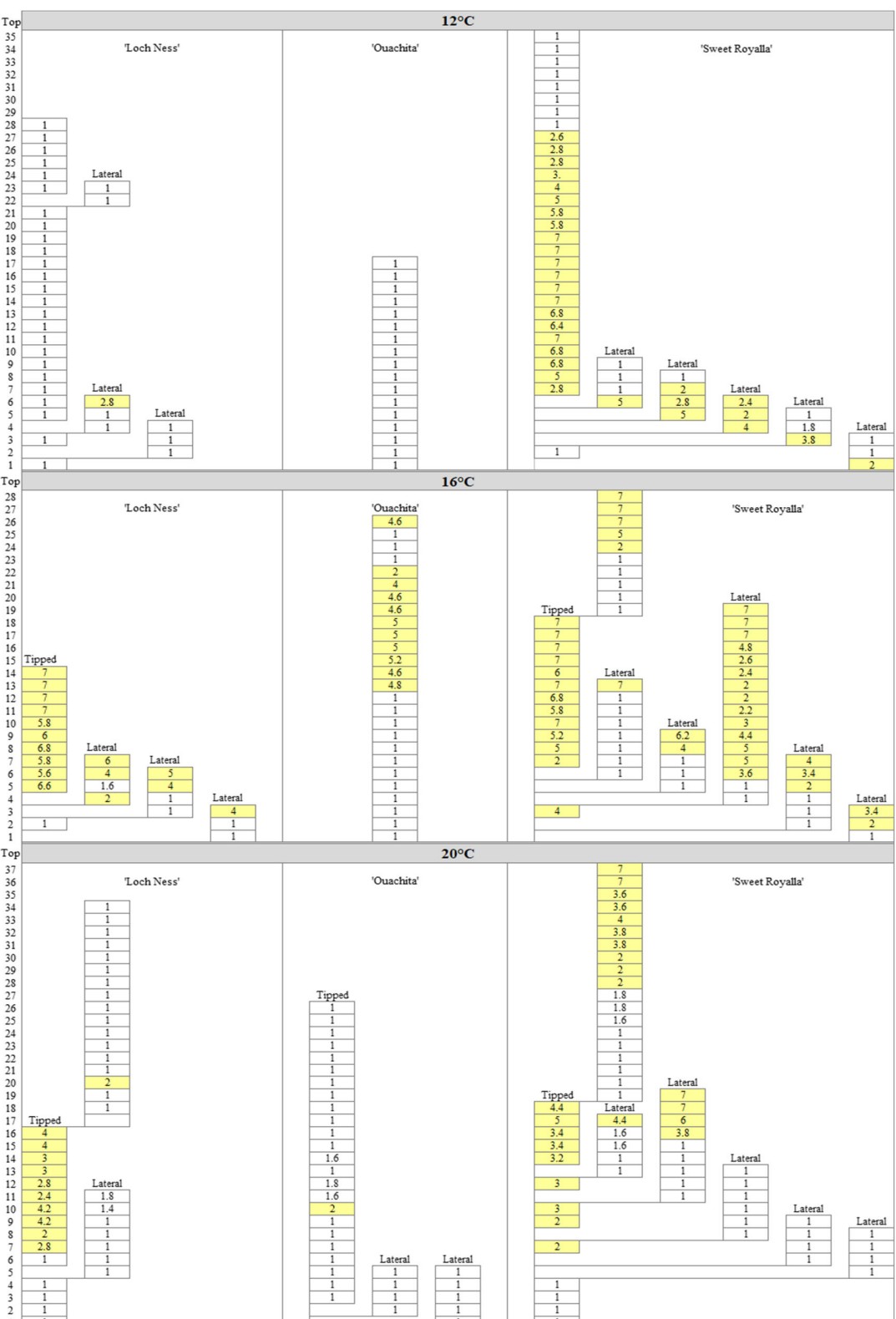

**Figure 2.** Profiles of flower development stages of lateral buds along the entire length of the main shoot as well as all lateral shoots of three blackberry cultivars after 11 weeks of cultivation under naturally decreasing autumn daylength at 60° N and temperatures of 12, 16 and 20 °C as indicated. Data are the results for a single plant of each cultivar and treatment. Yellow color denotes buds with flower stage ≥2, and the numbers included for each bud denote flower development stages according to the 1–7 stage scale. The numbers on the *y*-axis denote bud position along the shoots from base to top.

**Table 1.** Flowering time and performance of three blackberry cultivars in the spring under outdoor conditions as affected by temperature and naturally decreasing daylength in the previous autumn.

| Cultivar | Temperature (°C) | Days to Anthesis | Flowering Pants (%) | Flowers per Plant | Flowering Nodes (%) | Flowers per Lateral | Lateral Length (cm) |
|---|---|---|---|---|---|---|---|
| 'Loch Ness' | 12 | 37.8 cd | 100 | 168.2 cd | 60.2 b | 9.9 bc | 45.9 ab |
| | 16 | 34.4 cd | 100 | 138.0 d | 79.2 ab | 8.7 bc | 33.6 cde |
| | 20 | 38.4 cd | 100 | 222.2 bc | 91.6 a | 14.1 a | 40.9 bc |
| *Mean* | | *36.9* | *100* | *176.1* | *77.0* | *10.9* | *40.1* |
| 'Ouachita' | 12 | >100 a | 0 | 0.0 e | 0.0 d | 0.0 e | 25.3 e |
| | 16 | 74.2 b | 40 | 10.6 e | 12.7 d | 1.6 e | 26.2 de |
| | 20 | 40.8 c | 100 | 52.2 e | 36.6 c | 4.0 d | 49.4 a |
| *Mean* | | *71.7* | *46.7* | *20.9* | *14.4* | *1.9* | *33.6* |
| 'Sweet Royalla' | 12 | 33.4 cd | 100 | 226.0 ab | 76.7 ab | 8.6 c | 34.9 cd |
| | 16 | 23.8 d | 100 | 190.4 bc | 92.2 a | 10.9 b | 30.3 de |
| | 20 | 43.2 c | 100 | 274.0 a | 84.7 a | 8.6 c | 34.5 cde |
| *Mean* | | *33.5* | *100* | *230.1* | *84.5* | *9.4* | *33.2* |
| Probability levels of significance by ANOVA Source of variation | | | | | | | |
| Temperature (A) | | 0.009 | 0.003 | 0.02 | 0.003 | 0.001 | 0.01 |
| Cultivar (B) | | <0.001 | <0.001 | <0.001 | <0.001 | <0.001 | 0.007 |
| A × B | | <0.001 | <0.001 | n.s. | n.s. | 0.001 | <0.001 |

Values within the same column followed by different letters are significantly different at $p \leq 0.05$ by Tukey's test for the different temperatures and cultivars. The data are the means of five replicate plants.

## 4. Discussion

Unfortunately, we had to decapitate the vigorously growing plants as they became unmanageable at the higher temperatures after four weeks of cultivation. Since shoot tipping is shown to influence flowering in both biennial- and annual-fruiting blackberries [8,17], the results of the decapitated plants are not directly comparable with those of the non-decapitated plants at 12 °C. Accordingly, only tentative conclusions are permissible from comparison of these results.

Nevertheless, the observed timing and direction of flower development of the non-decapitated plants of 'Sweet Royalla' and 'Ouachita' presented in Figure 2, raise serious questions about the validity of the strict acropetal direction of flower development of buds within each cane of biennial-fruiting blackberry that was reported by Takeda et al. cf. [8,10]. Rather, the results concur with the results reported for a wide range of red raspberry [3–5,12,13], and clearly demonstrate that in both species, floral initiation starts in lateral buds located several nodes below the apex and then proceeds in both basipetal and acropetal directions. It was also found that when 'Sweet Royalla' was decapitated several nodes below the apex (in the region where FBI seems to start), the subtending bud did not grow out to form a shoot as usual but formed flower buds. This is an interesting finding that indicates that the first floral buds were initiated several buds below the apex, and furthermore, that these buds had been marginally induced to flower by 31 August when the plants were decapitated (cf. Figure 2).

The results in Figures 1 and 2 also demonstrate that in blackberry, there is no coincidence between cessation of growth and FBI as is the case in red raspberry cultivars [3–5,12,13]. On the contrary, 'Ouachita' which had a complete and early growth cessation at 12 °C, did not initiate any floral primordia at all at this temperature, whereas 'Sweet Royalla' had abundant FBI at all temperatures in the absence of growth cessation (Figures 1 and 2). The observed temperature effects on FBI under naturally decreasing autumn daylength is in full agreement with the well-known interaction of photoperiod and temperature in photoperiodic plants [20]. Commonly, the effect of photoperiod can be

greatly modified by temperature as also observed here. It is clear, however, that the temperature response varies greatly among the genetically highly diverse blackberry cultivars.

The results also show that under controlled temperature conditions, FBI of the three blackberry cultivars took place one to two months earlier in the season than previously reported for other cultivars under field conditions [9–12]. Thus, in controlled environment, floral initiation had taken place well before the treatments were terminated in mid-October, whereas under field conditions it was commonly reported to take place later in autumn or winter or not until spring [9–12]. In the present experiment, FBI was earliest at 16 °C in all tested cultivars, whereas no initiation took place in 'Ouachita' at a moderately low temperature of 12 °C, whereas in 'Lock Ness' the process was clearly delayed at this temperature (Figure 2). These results suggest that temperature is as important as short photoperiod for FBI in biennial fruiting blackberry. It is also interesting that winter chilling tends to enhance flowering and yield potential in marginally induced blackberry plants. This is an unusual flowering response, and it is particularly interesting that the same response was reported for red raspberry plants [21]. Thus, flowering and yield potential of potted long cane raspberry plants increased steadily as cold storage at −1 °C was extended from 4 to 12 and 20 weeks. Clearly, this is a flowering response that deserves further investigation.

As briefly mentioned in the Introduction, FBI in annual-fruiting blackberry cultivars seems to be unaffected by photoperiod, whereas temperature plays an important indirect role through its profound effect on the amount of vegetative growth that takes place before FBI [16].

The presented results demonstrate large genotypic differences in growth and FBI in biennial-fruiting blackberry, and that further experiments would be needed for elucidation of the effect of temperature and photoperiod and their interaction in the control of growth and FBI in biennial-fruiting blackberry. Such experiments should use non decapitated plants of several cultivars with contrasting growth and flowering characteristics and systematic variation of both temperature and photoperiod.

## 5. Conclusions

The results demonstrate that there is no functional linkage between cessation of growth and FBI in the blackberry as has been demonstrated for red raspberry. In contrast to previous reports, it is also clear that FBI starts in lateral buds located several nodes below the apex and then progresses in both basipetal and acropetal directions, just as in red raspberry. Furthermore, the results suggest that temperature is as important as photoperiod for the control of FBI in biennial-fruiting blackberry. However, because of the large genotypic variation among blackberry cultivars, further experiments with more cultivars and variation in both parameters are needed for clarification of this important issue.

**Author Contributions:** Conceptualization, A.S. and O.M.H.; methodology, A.S. and O.M.H.; formal analysis, A.S; investigation, A.S.; writing—original draft preparation, O.M.H.; writing—review and editing, A.S.; project administration, A.S.; funding acquisition, A.S. Both authors have read and agreed to the published version of the manuscript. All authors have read and agreed to the published version of the manuscript.

**Funding:** A.S. and O.M.H. acknowledge financial support to the research leading to these results from Project implemented under the Norwegian Financial Mechanism for 2014–2021 "We work together for a green, competitive and favorable social integration in Europe, contract number: NOR/POLNOR/QualityBerry/0014/2019-00", and the Norwegian Agricultural Agreement Research Fund/Foundation for Research Levy on Agricultural Products, grant number 326688 and Grofondet, grant number 200033.

**Data Availability Statement:** The raw data supporting the findings reported in this study are available on request from the corresponding author.

**Acknowledgments:** We acknowledge the skillful technical assistance of Rodmar Rivero and Kari Grønnerød, and Mirjana Sadojevic with plant management and bud sampling and dissection.

**Conflicts of Interest:** The authors declare no conflict of interest.

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
