# Peer review of "Temperature and Daylength Effects on Growth and Floral Initiation in Biennial-Fruiting Blackberry"

_horticulturae, doi:10.3390/horticulturae9121285_

Round 1

Reviewer 1 Report

Comments and Suggestions for Authors

Author Response

Dear Reviewer,

Thanks for the helpfull comments and suggestions for improvement of our manuscript. Below is our point-to-point responses to your raised questions and suggestions.

Line 105: Indicate the year in which the experiment was carried out!

A: The experiment was started in 2022 (see l. 109).

Line 107: For each variety studied, indicate the height of the shoots when the trial was started in the daylight phytotron. This is necessary because the shoots were decapitated at a height of about 350 cm.

A: This has been implemented as suggested on l. 110.

Line 109: Enter the number of plants per variety participating in the experiment.

A: As stated on l. 123 in the original version (l. 128 in the revised version), each treatment was started with six plants in each treatment.

Line 112: The term “flowering stages” is not appropriate, as flowering does not take place until the following year. The authors must use another suitable term, for example “stage of flower bud formation”. The same applies to line 128.

A: (l. 112/128) The wording has been amended to “flower development stage” (l. 133 - 134).

Line 132: Please indicate which parameters of flowering performance were included and explain their determination!

A: The information is given on l.132-135 in the original version (l. 142-144 in the revised version).

Line 137: Which test did the author use for the mean separation?

A: Tukey’s test was used for separation of means (See l. 147).

Line 147-152. All explanations of decapitation of shoots should be moved to the Materials and Methods section!

A: The transfer has been implemented as suggested (l. 138-142).

Line 151-153: The formation of new leaves is not possible without new shoot growth. Since the shoot tip has been removed, the buds below the decapitation point need some time to take over the function of the terminal bud. This statement is therefore very strange and needs to be better explained!

A: Rewording has been implemented as suggested on l. 157-158.

Line 156-158: This statement is superfluous, and it makes no sense to compare two fruit species with different growth potential.

A: Rewording has been implemented on l.163-164 as suggested.

(See also l. 170: The text has been reworded to “for a wide range of other cultivars.”).

Line 213: This statement is very unusual. The authors must explain how winter chill at 0 °C tend to enhance flowering in spring and have to find confirmation in the available literature.

A: Yes, this is an unusual flowering response which is hardly mentioned in the flowering physiology literature, and not at all for blackberries in which flowering is so little studied. However, it is highly interesting that the same response was reported for the genetically related red raspberry. Therefore, we have added a few sentences in the Discussion on l. 275-283 to discuss the phenomenon.

Line 214-215: This sentence is not true. Table 1 show that the percentage of flowering nodes in the varieties 'Loch Ness' and 'Sweet Royalla' at temperature 16 was not significantly different compared to the other two temperatures studied. The number of flowers per plant was also higher in the varieties studied at temperatures 12 and 20 than at 16 °C. The authors must explain this fact!

A: Unfortunately, there seems to be some misunderstanding here (confusion of the terms “advancement” and “enhancement”). We did not state here that flowering was enhanced, but only referred to a slight advancement of flowering (anthesis) in spring. In order to clarify the issue, we have reworded the text on lines 221-225 in the resubmitted version.

Line 219-221: In the cultivar ‘Ouachita’, the flowers per plant and the percentage of flowering nodes were higher or significantly higher at temperature 20 than at temperature 16. The results in Figure 2 are completely opposite. The authors have to explain how!

A: The reason for this apparent contradiction is that Figure 2 refers to the dissection results in autumn, while Table 1 refers to the final flowering results in spring. As stated above, the winter chilling did modify the flowering responses.

Line 282-283: In CONCLUSION the authors claim that the results suggest that temperature is as important as photoperiod for the control of FBI in biennial-fruiting blackberry. In my opinion, however, the results are rather confused and do not clearly show how the three different temperatures affect the FBI.

A: We hope that with the corrections and explanations given here and in the final text, the results will be less “confusing” and justify our conclusion that temperature is as important as photoperiod for the control of FBI in biennial-fruiting blackberry.

As the reviewer will notice, we have added two new references in the list.

Reviewer 2 Report

Comments and Suggestions for Authors

The authors propose an interesting study regarding the influence of temperature on growth and floral initiation in biennial-fruiting blackberry. In order to do so, the authors used a phytotron, which allowed them to control growth conditions. However I have found several important problems that minimize the impact of the experiment.

Line 2 Firstly, the scientific name of the species should be addressed, if not in the title, at least in the introduction.

Line 27 The introduction is very limited in regard to the references cited and is slightly chaotic. The only clear point is that floral initiation in blackberry is highly dependent on the cultivar. Given this, the authors should have explained the criteria used to choose the cultivars for the experiments and justified why only three of them were sufficient for this experiment.

Line 109: Related to the previous point, the authors do not provide information or justification for the temperatures selected for the study. This aspect should be addressed to clarify the most crucial aspect of the experimental design.

Line 127 Another important point is the dissection of a single plant per treatment as a method to study the flowering stage. Drawing conclusions from a single repetition is not advisable, and in any case, should be used as a model to compare with the results recorded in spring. The sample size should have been larger to make meaningful comments on the phenological states.

Line 149 The decapitation of some plants and not all of them made the results incomparable between treatments. However, in Table 1, the ANOVA showed that the analysis was done without discriminating this effect. This is a significant error that hampers a meaningful interpretation of the results.

Line 231 The discussion is very short and the references are almost non-existent. The authors should try to expand it more and introduce literature relevant to the theme of the manuscript

Line 278 The results obtained do not support their initial hypothesis, since the mechanisms that could explain the effect of temperature remain completely unknown. Maybe the authors should consider shifting the focus of the manuscript to the relationship between vegetative growth and floral bud initiation.

Author Response

Dear Referee

Thanks for your helpful comments and suggested corrections. We have adapted the manuscript acordingly. Below, please find our point-to-point responses to each of your raised questiond.

Line 2 Firstly, the scientific name of the species should be addressed, if not in the title, at least in the introduction.

A: The “scientific name” and its status are given in the original manuscript on l. 40-41.

Line 27 The introduction is very limited in regard to the references cited and is slightly chaotic. The only clear point is that floral initiation in blackberry is highly dependent on the cultivar. Given this, the authors should have explained the criteria used to choose the cultivars for the experiments and justified why only three of them were sufficient for this experiment.

A: The simple explanation for the limited number of references is that so little research has been done on the physiology of flowering in blackberries. We cannot cite references that do not exist. However, two highly relevant references have now been added to the list. The justification for the selection of cultivars for the experiment is now added on L. 100 and 105- 106.

Line 109: Related to the previous point, the authors do not provide information or justification for the temperatures selected for the study. This aspect should be addressed to clarify the most crucial aspect of the experimental design.

A: The justification for the selected temperatures is now added on L. 114-115. We believe that the 12 to 20 ℃ temperature range is highly relevant for the autumn environment in which biennial-fruiting blackberry is commonly grown.

Line 127 Another important point is the dissection of a single plant per treatment as a method to study the flowering stage. Drawing conclusions from a single repetition is not advisable, and in any case, should be used as a model to compare with the results recorded in spring. The sample size should have been larger to make meaningful comments on the phenological states.

A: The number of replications that can be accommodated in controlled environment experiments such as this, and especially with bulky blackberry plants, will have to be limited. It should also be kept in mind that the dissection of all buds along the entire length of nine plants is a tedious and time-consuming process that involves several hundred buds. Therefore, starting with six replicate plants, we decided to place the main emphasis on the flowering performance in spring with five replications. Just as the Ref. 2 suggest, the bud dissections were mainly used as “a model” for comparison with the spring flowering. We also think that the results presented in Figure 2, provide an interesting picture of the plants’ architecture and the position and number of initiated buds.

Line 149 The decapitation of some plants and not all of them made the results incomparable between treatments. However, in Table 1, the ANOVA showed that the analysis was done without discriminating this effect. This is a significant error that hampers a meaningful interpretation of the results.

A: The text of our discussion (l. 240-245) demonstrates that we are fully aware of the problems involved with decapitation of some plants, clearly stating that only tentative conclusions are permissible from such data. Nevertheless, we have deliberately performed an ANOVA of the full dataset to get at least some information about the degree of variance. We hope that this is permissible as long as cautions are taken about the conclusions.

Line 231 The discussion is very short, and the references are almost non-existent. The authors should try to expand it more and introduce literature relevant to the theme of the manuscript.

A: See the above comments under l. 27. However, we have cited two new, and highly relevant references in the resubmitted version.

Line 278 The results obtained do not support their initial hypothesis, since the mechanisms that could explain the effect of temperature remain completely unknown. Maybe the authors should consider shifting the focus of the manuscript to the relationship between vegetative growth and floral bud initiation.

A: We have not presented any initial hypothesis, but simply stated that we wanted to “study the effect of temperature on growth and FBI in cultivars with contrasting growth characteristics” (see l. 98-99 in the original manuscript). However, in our revised version we have added a short statement and added an additional reference to explain the possible effect of temperature on flowering (l. 264-268 in the revised manuscript). Thus, it is well known that temperature can strongly modify the effect of daylength on photoperiodic plants, and it’s clear that this is what happened in our experiment.

Thanks, and regards

Reviewer 3 Report

Comments and Suggestions for Authors

A simple experiment was carried out, it is not known when (year). Why wasn't this experiment repeated with or without pruning the apical shoots at all temperature levels. Percentage values of changes due to temperature and comparison of varieties should be provided. In the discussion, provide details about the results of other publications to which the current results are compared. All important comments are marked in the text of the publication.

Comments on the Quality of English Language

Good level of English.

Author Response

Dear Reviewer,

Thanks for your helpful comments and suggested corrections to improve our manuscript. Please find our responses below, and in the attached revised manuscript.

A simple experiment was carried out, it is not known when (year). Why wasn't this experiment repeated with or without pruning the apical shoots at all temperature levels. Percentage values of changes due to temperature and comparison of varieties should be provided. In the discussion, provide details about the results of other publications to which the current results are compared. All important comments are marked in the text of the publication.

A: The reviewer wonders why we haven’t repeated the experiment with and without pruning. That is exactly what we recommend for further research in our conclusion. However, it should be kept in mind that this type of experiment is not so quickly replicated since it takes a full growing season.

In factorial experiments like this, it is the trend of the results that is most important rather than the specific percentage of change between individual treatments. Since the present case involves three cultivars and three temperatures, it would also be a bit pedantic to provide al the values.

We have also adapted and corrected the text throughout the manuscript, according to the helpful suggestions from you (please see the attached revised manuscript).

Thanks, and kind regards

Round 2

Reviewer 1 Report

Comments and Suggestions for Authors I have carefully reviewed the new version of the manuscript entitled “Temperature and daylength effects
on growth and floral initiation in biennial-fruiting blackberry”. Since important misunderstandings from the
previous version of article were clarified I don’t have any significant remark to new version of manuscript.

Author Response

I have carefully reviewed the new version of the manuscript entitled “Temperature and daylength effects on growth and floral initiation in biennial-fruiting blackberry”. Since important misunderstandings from the previous version of article were clarified I don’t have any significant remark to new version of manuscript.

Dear Reviewer 1,

We are grateful for all help for improvement of the manuscript, and appreciate your acceptance of the current version of the manuscript.

Kind regards,

Authors

Reviewer 2 Report

Comments and Suggestions for Authors

A: The number of replications that can be accommodated in controlled environment experiments such as this, and especially with bulky blackberry plants, will have to be limited. It should also be kept in mind that the dissection of all buds along the entire length of nine plants is a tedious and time-consuming process that involves several hundred buds. Therefore, starting with six replicate plants, we decided to place the main emphasis on the flowering performance in spring with five replications. Just as the Ref. 2 suggest, the bud dissections were mainly used as “a model” for comparison with the spring flowering. We also think that the results presented in Figure 2, provide an interesting picture of the plants’ architecture and the position and number of initiated buds.

R: The experimental design must be adapted to the environment, of course. A single more replicate would have given the opportunity to have three replicates for the dissection of buds and three replicate for spring flowering, which would have provided a better perspective and much more solid results. This would have enhanced the robustness of the study and have provided a more comprehensive understanding of the plant's response. I want to remind the authors, for future experiments, that the tedious and time-consuming work is also an essential part of our job as researchers

A: The text of our discussion (l. 240-245) demonstrates that we are fully aware of the problems involved with decapitation of some plants, clearly stating that only tentative conclusions are permissible from such data. Nevertheless, we have deliberately performed an ANOVA of the full dataset to get at least some information about the degree of variance. We hope that this is permissible as long as cautions are taken about the conclusions.

R: Additional analyses or incorporating factors related to decapitation might have provided a more nuanced interpretation of the results. Even so, the conclusions that can be taken are scarce. In my opinion, the statistical analysis should be changed before publication.

A: We have not presented any initial hypothesis, but simply stated that we wanted to “study the effect of temperature on growth and FBI in cultivars with contrasting growth characteristics” (see l. 98-99 in the original manuscript). However, in our revised version we have added a short statement and added an additional reference to explain the possible effect of temperature on flowering (l. 264-268 in the revised manuscript). Thus, it is well known that temperature can strongly modify the effect of daylength on photoperiodic plants, and it’s clear that this is what happened in our experiment.

R: For future research, I would advise the researcher to follow the scientific method and formulate hypothesis before setting up the experiments. Clearly defined hypotheses can guide the research process, improve the focus of the study, and contribute to more meaningful interpretations.

Best regards

Author Response

Dear Reviewer 2,

R: The experimental design must be adapted to the environment, of course. A single more replicate would have given the opportunity to have three replicates for the dissection of buds and three replicate for spring flowering, which would have provided a better perspective and much more solid results. This would have enhanced the robustness of the study and have provided a more comprehensive understanding of the plant's response. I want to remind the authors, for future experiments, that the tedious and time-consuming work is also an essential part of our job as researchers.

A: We fully agree with your weighty and critical comments. However, at the present state there is no possibility to re-do our unfortunate choice of the number of replicate plants for dissection.

R: Additional analyses or incorporating factors related to decapitation might have provided a more nuanced interpretation of the results. Even so, the conclusions that can be taken are scarce. In my opinion, the statistical analysis should be changed before publication.

A: In principle, we agree with your comment. However, we do not quite see how the statistical analysis could be re-done. For example, for two of the cultivars, we tipped all plants at the two highest temperatures, whereas for ‘Ouachita’, which completely ceased growing at both 12 and 16°C, only the plants at 20°C were tipped. Nor do we think that a new analysis would improve the interpretation and conclusion of the results. Therefore, we prefer to keep Table 1 as it stands, since it gives at least some information about the degree of variance involved.

R: For future research, I would advise the researcher to follow the scientific method and formulate hypothesis before setting up the experiments. Clearly defined hypotheses can guide the research process, improve the focus of the study, and contribute to more meaningful interpretations.

A: We appreciate the comment and have taken notice of your advice for future research.

Kind regards,

Authors